# Insights into magma ocean dynamics from the transport properties of basaltic melt

Suraj K. Bajgain [1,2] ✉, Aaron Wolfgang Ashley[1], Mainak Mookherjee [1] ✉, Dipta B. Ghosh[3] & Bijaya B. Karki [3] ✉

The viscosity of magma plays a crucial role in the dynamics of the Earth: from the crystallization of a magma ocean during its initial stages to modern-day volcanic processes. However, the pressure-dependence behavior of viscosity at high pressure remains controversial. In this study, we report the results of first-principles molecular dynamics simulations of basaltic melt to show that the melt viscosity increases upon compression along each isotherm for the entire lower mantle after showing minima at ~6 GPa. However, elevated temperatures of the magma ocean translate to a narrow range of viscosity, i.e., 0.01–0.03 Pa.s. This low viscosity implies that the crystallization of the magma ocean could be complete within a few million years. These results also suggest that the crystallization of the magma ocean is likely to be fractional, thus supporting the hypothesis that present-day mantle heterogeneities could have been generated during the early crystallization of the primitive mantle.

Silicate melts play a key role in the terrestrial planet by causing chemical differentiation. In the early history of the Earth, silicate melts would have strongly influenced mantle dynamics because of violent collisions that resulted in large-scale melting of the mantle, i.e., magma ocean stage[1–4]. If such a magma ocean existed, dense metallic melt droplets would likely settle through the silicate-rich molten mantle[5,6]. The viscosity of such a magma ocean or its constituent silicate melt is crucial for determining the thermal and chemical evolution of the planet. This is because the melt viscosity directly influences the cooling time of the magma ocean, the nature of its crystallization, melt percolation, and the rate at which the metal droplets could sink in the magma ocean[7–10]. Silicate melt viscosity also affects magma transport in the modern mantle as silicate melts are likely to be present in various tectonic settings across the entire mantle including the base of the lower mantle[11–15]. Most of the crustal material on Earth is the product of partial melting of the mantle that produces basaltic melts[16,17]. Basaltic melts are produced mostly by decompression melting along mid-oceanic ridges in the upper mantle. However, both geophysical and geochemical evidence also suggests the presence of partial melts, likely basaltic in composition, in the deep mantle above the core-

mantle boundary (CMB)[13,18–22]. Such a melt layer above CMB may represent a remnant fraction of the magma ocean and thus may have implications for the origin and preservation of geochemical signature in the mantle[23–25].

Despite its significance, experiments on the viscosity of silicate melts are mostly available for pure end member compositions[26–30], and experimental constraints are often limited to 25 GPa owing to technical challenges at the high pressures relevant for the magma ocean[10,31–34]. The pressure dependence of viscosity is sensitive to the melt composition. Yet, there are disagreements between studies even for the same melt compositions[10,26,31,35,36]. Extrapolations of viscosity from experiments conducted at relatively low-pressures to higher pressures often add significant uncertainties due to pressure-induced changes to the atomistic scale structure of the melt and resulting changes in properties. For instance, the Si−O coordination often increases from 4-fold to 6-fold upon compression from ambient pressures to high pressure[37–39]. Changes in the atomic scale structure of melts are likely to exert a significant influence on their macroscopic properties such as viscosity. First-principles molecular dynamics (FPMD) simulations, which are complementary to experiments, have proven to be

[1]Earth Materials Laboratory, Earth, Ocean and Atmospheric Sciences, Florida State University, Tallahassee, FL, USA. [2]Department of Geology, School of Natural Resources & Environment, Lake Superior State University, Sault Ste Marie, MI, USA. [3]School of Electrical Engineering and Computer Science, Department of Geology and Geophysics, Center for Computation and Technology, Louisiana State University, Baton Rouge, LA, USA. ✉e-mail: sbajgain@lssu.edu; mmookherjee@fsu.edu; bbkarki@lsu.edu

extremely useful in simultaneously evaluating structure, elasticity, and transport properties including melt viscosity at high pressure and temperature conditions[38,40–45]. However, prior FPMD simulations on basaltic melts show divergent results, one of the studies showed that the viscosity increases with increasing pressure at pressure ≥10 GPa (i.e., depths > 300 km)[44] whereas another study showed a significant reduction in the viscosity at pressures ≥40 GPa at temperatures 2200–3000 K[46]. Since the timescale of freezing of the magma ocean depends largely on the viscosity of the magma ocean, it is important to have better constraints on the viscosity of its constituent silicate melt[3,47,48]. Even though the studies based on isotope geochemistry show evidence of heterogeneity in the Earth's lower mantle, the sources of heterogeneity are poorly constrained because of limited information on the crystallization of the magma ocean[10,30]. Fractional solidification of the magma ocean would suggest that observed chemical heterogeneity in the present-day Earth's mantle was preserved from early magma ocean solidification[49,50]. In contrast, equilibrium solidification would imply that the chemical heterogeneity was added at a later stage of the Earth's history[51]. Melt viscosity is likely to have a significant influence on how the magma ocean solidified[10,30]. Therefore, a better constraint on the viscosity of the magma ocean is crucial for understanding present-day mantle heterogeneity.

Here, we estimate the viscosity of basaltic melt using first-principles simulations over the entire mantle pressure-temperature regime (0–136 GPa and 2200–4000 K). This work provides strong constraints on the viscosity of the magma ocean given the important implication of melt viscosity on the magma ocean dynamics and pla-netary differentiation. Our results on the structure and properties of mafic melts indicate that the magma ocean is likely to have cooled in less than a few million years with fractional crystallization.

## Results and discussion
### Thermal equation of state
We find that densities estimated using FPMD agree well with available experimental data from X-ray diffraction[37], and sink-float[52,53] methods. Slight discrepancies can be attributed mainly to variations in melt chemistry (Fig. 1). The calculated pressure as a function of volume (or density) along all three isotherms: 4000 K, 3000 K, and 2200 K can be described using the Mie-Grüneisen thermal equation of state:

$$P_{V,T} = P_{V,T_{ref}} + \left(\frac{dP}{dT}\right)_V (T - T_{ref}) \tag{1}$$

Here $P_{V,T_{ref}}$ is the pressure along a reference isotherm (2200 K) which can be represented with the third-order Birch Murnaghan equation of state (Table 1). The $\left(\frac{dP}{dT}\right)_V$ term is the thermal pressure at a constant volume. We find that $\left(\frac{dP}{dT}\right)_V$ changes by an order of magnitude when melt volume is compressed to half of the reference volume ($V_{ref}$ = 3422.47 Å³). The density at $V_{ref}$ is 2.55 g cm⁻³. This indicates that the thermal pressure increases with decreasing volume or increasing density (Fig. 1). In our analysis, we do not include the data that lie outside of the molten $P$–$T$ regime. At the $P$–$T$ conditions corresponding to the solid (glassy) regime, the calculated densities are lower compared to the prediction based on the equation of state. Along all

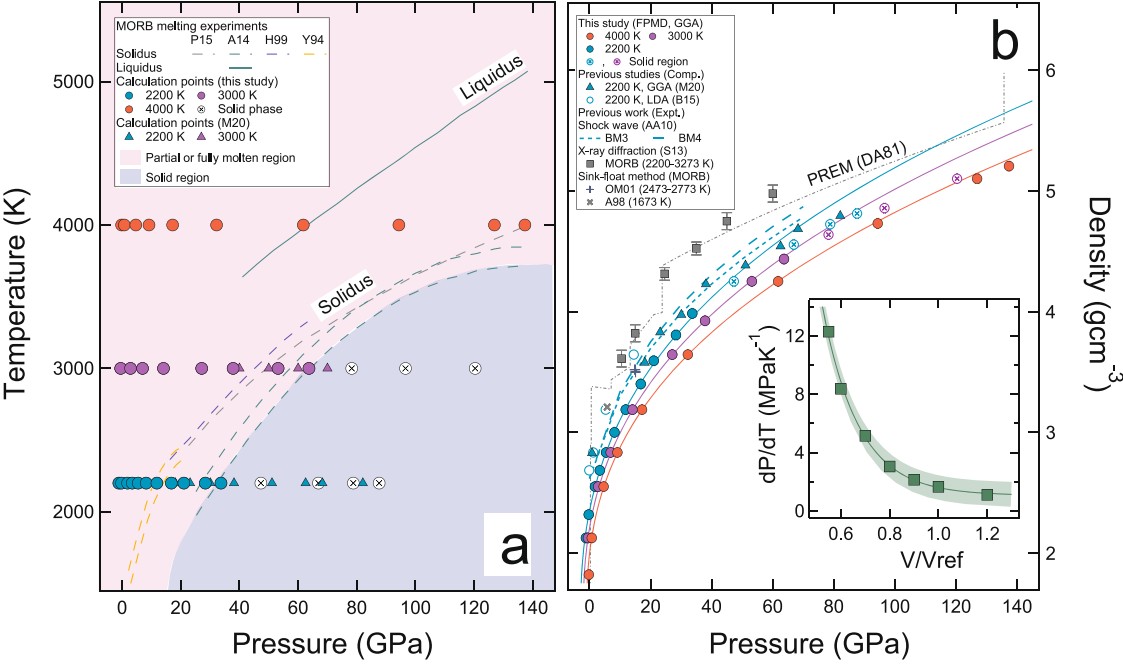

**Fig. 1 | Melting temperature and equation of state. a** Thermodynamic states of data points in this study along with the melting curve of basaltic melt from previous work for reference. Light pink and light purple shaded regions in the figure represent the $P$–$T$ space where melt and solid phases are stable, respectively. Dashed and solid lines represent MORB solidus[71,78–80] and liquidus[71], respectively, from previous experiments. Filled circles show the temperature and pressure conditions of the simulation data points that are within the liquid or partial melt region of the phase diagram. Open circles with crosses are the data points that are in the solid regime of the phase diagram and thus not used in the analysis of the results. For comparison, the calculation points from the previous FPMD study at 3000 K and 2200 K are also shown by upward pointing triangles[46]. **b** Calculated pressure-density results (filled circles) of the model basaltic melt at 4000 K (red),

3000 K (purple), and 2200 K (cyan) with the equation of state isotherms using Eq. 1 (solid curves). Dashed lines are the 3rd and 4th order Birch Murnaghan thermal equation of state representations of shock wave experiments of model basalt melt[54]. For comparison, the experimental data for MORB melts using the X-ray diffraction method (S13)[37] and sink-float method (A98, OM01)[52,53] are also shown with filled gray squares and gray pluses/crosses, respectively. Open circles with crosses are the data points that were not used in the thermal equation of state (Eq. 1) analysis. The inset in panel (**b**) represents the calculated (symbols) and model (curve) results for thermal pressure coefficient $\left(\frac{dP}{dT}\right)_V$ as a function of volume ratio ($V_{ref}$ = 3422.47 Å³) and defined by $\left(\frac{dP}{dT}\right)_V = 0.89 + 291.92 e^{-6.04 \times u}$. Green shaded band shows the uncertainties with ±1σ standard deviation. Errors in pressures (±1σ uncertainties) are smaller than the symbol size.

**Table 1 | Density ($\rho_0$), bulk modulus ($K_0$), and the pressure derivatives of bulk modulus ($K'_0$) for silicate melts**

| Composition | T (K) | $\rho_0$ (g cm$^{-3}$) | $K_0$ (GPa) | $K'_0$ | $K''_0$ | Fit type | Method | References |
|---|---|---|---|---|---|---|---|---|
| MB | 2200 | 2.37 ± 0.02 | 21.9 ± 2.9 | 3.79 ± 0.38 | −0.13 ± 0.05 | BM4 | FPMD, GGA | This study |
|  |  | 2.35 ± 0.02 | 18.4 ± 1.7 | 4.70 ± 0.22 |  | BM3 | FPMD, GGA | This study |
|  |  | 2.70 ± 0.04 | 19.4 ± 4.2 | 6.20 ± 1.57 |  | BM3 | FPMD, LDA | [a]Bajgain et al. (2015)[38] |
| MB | 3000 | 2.25 ± 0.02 | 18.6 ± 1.6 | 4.02 ± 0.14 | −0.18 ± 0.03 | BM4 | FPMD, GGA | This study |
|  |  | 2.18 ± 0.02 | 12.4 ± 1.1 | 5.68 ± 0.18 |  | BM3 | FPMD, GGA | This study |
|  |  | 2.61 ± 0.02 | 27 ± 2 | 3.7 ± 0.2 | 0.06 ± 0.01 | BM4 | FPMD, LDA | Bajgain et al. (2015)[38] |
|  |  | 2.59 ± 0.14 | 19.8 ± 9.8 | 5.97 ± 1.33 |  | BM3 | FPMD, LDA | [a]Bajgain et al. (2015)[38] |
| MB | 4000 | 1.91 ± 0.02 | 6.83 ± 0.72 | 5.07 ± 0.25 | −0.86 ± 0.21 | BM4 | FPMD, GGA | This study |
|  |  | 1.90 ± 0.02 | 6.08 ± 0.55 | 6.95 ± 0.23 |  | BM3 | FPMD, GGA | This study |
|  |  | 2.38 ± 0.08 | 15.12 ± 3.75 | 6.11 ± 0.49 |  | BM3 | FPMD, LDA | [a]Bajgain et al. (2015)[38] |
| MB | 1673 | 2.62[b] | 22.98[c] | 4.66 | −0.149 | BM4 | Shock wave | Asimow and Aherns (2010)[54] |
|  |  | 2.62[b] | 22.98[c] | 5.36 |  | BM3 |  |  |
| MB | 2273 | 2.54 ± 0.10 |  |  |  |  | MD classical | Vuilleumier et al. (2009)[90] |
| MORB | 3000 | 2.7 ± 0.03 | 23 ± 1 | 3.6 ± 0.3 | 0.12 ± 0.04 | BM4 | FPMD, LDA | Bajgain et al. (2015)[38] |
| MORB | 2735 | 2.48[d] | 24 ± 1.7 | 0.66 ± 0.032 | −0.057 ± 0.0057 | BM4 | X-ray diffraction | Sanloup et al. (2013)[37] |
| MORB | 1673 | 2.65 | 20.5 | 5.2 |  | BM3 | MD classical | Dufils et al. (2017)[43] |
| Pyrolite | 3000 | 2.26 ± 0.02 | 14 ± 1 | 5.7 ± 0.2 |  | BM3 | FPMD, GGA | Solomatova and Caracas (2021)[104] |
|  | 4000 | 2.08 ± 0.03 | 12 ± 1 | 5.4 ± 0.1 |  |  |  |  |

*MB* Model basalt (36 mol % anorthite and 64 mol% diopside).
[a]Refitted using LDA data from Bajgain et al. (2015).
[b]Density adopted from partial molar volume of oxides (Lange, 1997).
[c]Adopted from Ali and Lange, 2008 (fixed).
[d]Density was fixed.

explored isotherms, basaltic melts are highly compressible at low pressure, and they become stiffer at higher pressures (Fig. 1). The equation-of-state parameters compare well with experimental studies[37,54] (Table 1). Our zero pressure density ($\rho_0$) results using GGA are smaller than the LDA-based results[38] (Table 1). For example, the $\rho_0^{GGA}$ = 2.70 ± 0.04 g cm$^{-3}$ at 2200 K is ~13% lower than the $\rho_0^{LDA}$. The $\rho_0^{GGA}$ and $\rho_0^{LDA}$ tend to bracket melt densities from shock wave experiments[54,55], i.e., $\rho_0^{GGA} < \rho_0^{shock} < \rho_0^{LDA}$ (Fig. 1). This is consistent with the systematic difference between LDA and GGA, which has been well documented in silicate melts[45,56]. Along the 2200 K isotherm and up to 35 GPa, pressures calculated in our study are larger than the recent FPMD simulation which used the GGA method[46]. However, the difference in pressures reduces at higher densities (Fig. 1). The silicate melts are more compressible than mantle minerals whose zero pressure bulk moduli ($K_0$) are of the order of ~100 GPa and bulk moduli could reach a few hundred gigapascals at lower mantle conditions[57,58]. In comparison, the $K_0$ of model basaltic melt at 2200 K is ~20 GPa. The rapid increase in the density of silicate melt with increasing pressure is related to pressure-dependent changes in the atomic scale structure of the silicate melt (Supplementary Note 1, Figs. S1–S3).

## Transport properties

At high-pressure conditions relevant to the lower mantle, the mobility of atoms is significantly reduced due to diminished free volume, and consequently, the relaxation time increases. This means a considerably longer simulation time is required to obtain statistically converged transport properties of silicate liquids. We analyze the time evolution of the mean square displacement (MSD) of atoms and the stress auto-correlation function (ACF) to ascertain statistically meaningful convergence for all the P–T conditions explored in this study before evaluating the transport coefficients ("Methods" section). A linear relation between MSD and simulation time ensures that the atoms have reached a diffusive regime and the predicted transport coefficients in the diffusive regime should be statistically reliable (Fig. 2).

At the highest temperature of 4000 K explored in this study, the melt shows the diffusive regime, i.e., an MSD slope of unity within a few picoseconds of the simulation owing to the higher kinetic energies of the atoms. However, along relatively low-temperature isotherms, it takes a significantly longer time to reach the diffusive regime in the basaltic melt. It also takes a longer time for MSD to reach the diffusive regime at higher pressure due to diminished free volume in the melt for ion migration. An additional important criterion for a statistically meaningful result is to ensure that MSD exceeds 10 Å$^2$ which indicates that the atoms have moved at least double the average bond distances thus implying the ergodic behavior of melt[44,45,59]. Along 3000 K isotherm at pressures >60 GPa, the MSD does not reach a diffusive regime or exceed 10 Å$^2$ even after unusually long simulation runs exceeding 300 ps. Along the 2200 K isotherm, the MSD-time plot indicates that melt dynamics does not reach a diffusive regime at pressures >34 GPa (Fig. 2). Our long simulation results compare well with previous LDA simulations of the basaltic melt which showed MSD didn't reach a diffusive regime at 3000 K and 70 GPa after simulation times >100 ps and viscosity at 3000 K was only reported for pressure ≤40 GPa[44]. Transport coefficients that are estimated without statistical convergence often exhibit large uncertainties. Therefore, for a meaningful interpretation of our results, we discard the calculated transport coefficients at those P–T conditions that are subsolidus since it is likely that the basalt is in or approaching the glassy state (Fig. 1).

To predict the shear viscosity, we use the Green-Kubo relation ("Methods" section) and determine the integral values of the shear stress autocorrelation function (ACF)[60]. Since the ACF decays to zero within the timescale of a simulation, the Maxwell relaxation time is much shorter than seismic periods over the entire mantle regime[61]. This is an indication that seismic wave propagation in silicate melts at depth will occur in the relaxed limit. For the basaltic melt at high temperatures and low pressures (<15 GPa at 3000 K, <32 GPa at 4000 K), ACF converges to zero within a few picoseconds and fluctuates around zero thereafter due to the high mobility of basaltic melt (Fig. 2). However, it often requires a longer time for the ACF to

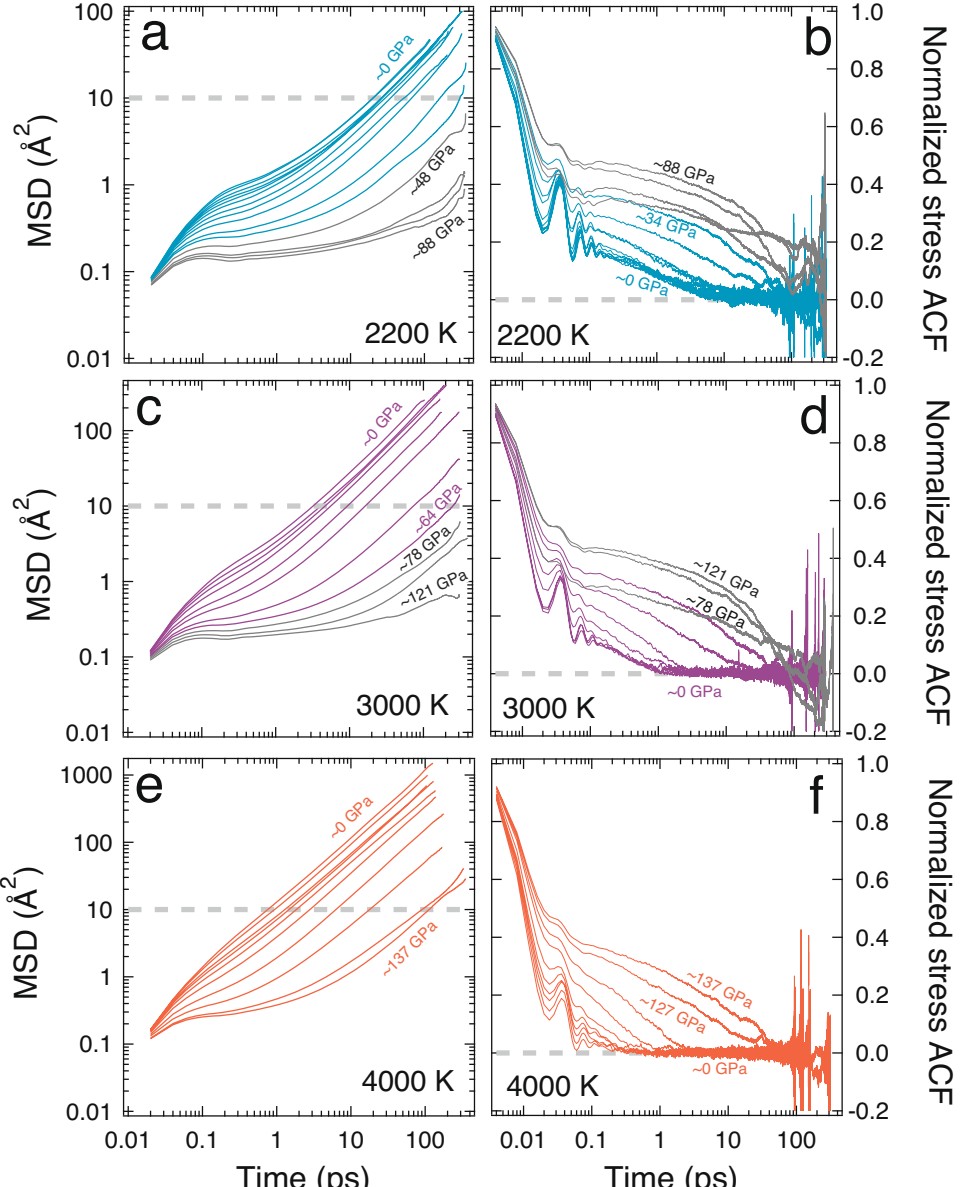

**Fig. 2 | Mean square displacement and stress autocorrelation function of basaltic melt.** Time evolution of mean square displacement (MSD) using Eq. 3 and normalized stress autocorrelation function (ACF) using Eq. 4 at 2200 K (**a** and **b**), 3000 K (**c** and **d**), and 4000 K (**e** and **f**). In each MSD plot, the gray dashed lines indicate that the MSD did not exceed 10 Å². Similarly, gray dashed lines in each ACF plot highlight the cases where ACF does not decay to zero. These gray lines show that the low-temperature simulations at high pressures do not achieve acceptable convergences even after ~350 ps of simulation.

converge at lower temperatures and at higher pressures. For instance, at 2200 K and ~0 GPa, ACF decays to zero only after ~10 ps of simulation time. Yet, the ACF at high pressures along any explored isotherm does not converge within our simulation timescales.

In basaltic melt, the self-diffusion coefficients of cations Ca and Mg are larger compared to the self-diffusion coefficients of cations Al and Si, and anion O along each explored isotherm (Supplementary Fig. S4). Magnesium is the fastest species followed by calcium and silicon is the slowest species. The diffusivities follow the order $D_{Mg} > D_{Ca} > D_O > D_{Al} > D_{Si}$. Self-diffusion coefficients of Ca and Mg both decrease with increasing pressure along all explored isotherms. However, the diffusivity of Al, Si, and O at the lower temperature of 2200 K show positive pressure dependence at lower pressure up to ~8 GPa. This behavior is similar to previous work on polymerized aluminosilicate melt[56,62,63] as well as depolymerized melt such as MgSiO₃ at lower temperatures[64]. A comparison of our results with previous LDA results

shows that the self-diffusion coefficients of all species are almost insensitive to the choice of the exchange correlation functional. Compared to the recent GGA simulations[46], our diffusivity results are slightly higher but follow a similar trend along 2200 K isotherm but we do not find an increase in diffusivity at ~60 GPa and 3000 K (Supplementary Fig. 4).

We find that the pressure dependence of viscosity for the basaltic melt is also sensitive to temperature. Along the 2200 K isotherm, the viscosity decreases with increasing pressure up to ~6 GPa (Fig. 3). Beyond this pressure, the viscosity begins to increase with continued compression. Yet, at temperatures ≥3000 K, we do not notice any negative pressure dependence of viscosity, i.e., the viscosity generally increases upon compression (Fig. 3). At 3000 K, viscosity increases gradually at lower pressures up to ~7 GPa and then increases more rapidly with further compression. Along the highest temperature explored in this study, i.e., 4000 K, viscosity increases more rapidly

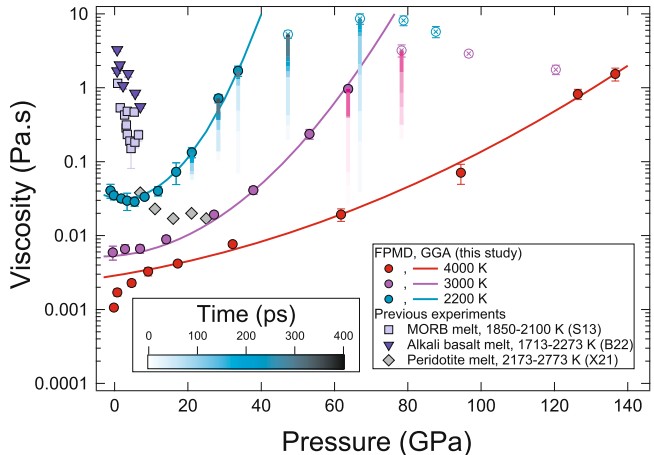

**Fig. 3 | Melt viscosity as a function of pressure.** The viscosity of model basaltic melt along different isotherms: 4000 K (filled red circles), 3000 K (filled purple circles), and 2200 K (filled cyan circles) as a function of pressure (this study, GGA). The solid curves represent a modified VFT model (Eq. 2) at 4000 K (red), 3000 K (purple), and 2200 K (cyan). Open circles with crosses are the calculated results that are not converged. Blue and purple shaded vertical lines in selected pressures show the viscosity calculation using various simulation times (in picoseconds) as indicated in the color bar in the figure. Filled purple squares, blue filled downward facing triangles, and filled yellow diamonds are the experimental viscosity data for multi-component silicate melts: MORB (S13)[31], alkali basalt (B22)[34], and peridotite (X21)[10], respectively. Error in viscosities represents ±1σ uncertainties.

upon compression at lower pressures. However, the effect of pressure on viscosity at high pressures becomes stronger along low-temperature isotherms.

For the entire mantle pressure range of 0–137 GPa and temperatures explored in this study (2200–4000 K), we use the modified Vogel-Fulcher-Tammann (VFT) equation[65] (Eq. 2) to model the combined pressure and temperature dependence of viscosity:

$$\eta_{P,T} = \exp\left[A + \frac{B}{(T - T_{VFT})}\right] \quad (2)$$

where $A = a + bP + cP^2$ and $B = d + eP + fP^2$ with $a = -6.8 \pm 0.2$, $b = 0.03 \pm 0.006$, $c = -0.0012 \pm 0.00013$, $d = 2241 \pm 621$, $e = -33.5 \pm 15$, $f = 3.5 \pm 0.5$, and $T_{VFT} = 1527 \pm 121$ K. The decrease in viscosity with compression at low pressures and low temperatures has also been reported in other polymerized silicate melts in prior computational[43,45,56,66–68] and experimental[29,31,36] studies (Fig. 3). We find that the minima in the anomalous pressure dependence of viscosity depend on the temperature and the degree of polymerization of silicate melts. Studies on aluminosilicate melts indicated that such minima were at ~8 GPa at 2500 K which is relatively higher than typical experiments[56]. Most of the experimental studies that are confined to temperatures below 2100 K show the minima in the anomalous pressure dependence of viscosity around 5 GPa[29,31,36]. Although prior experiments on depolymerized melts indicated a continuous increase of viscosity with increasing pressure, more recent experiments reported a reduction in viscosity with increasing pressure in the depolymerized melts with pyroxene and peridotite compositions[10,26,35].

Our FPMD results compare well with the previously predicted viscosity of the basaltic, diopside, and anorthite melts (Supplementary Fig. S5)[44,69,70]. Some discrepancies are likely because of differences in exchange-correlation functionals. Most prior studies have used LDA for electron exchange-correlation[44,69,70]. In this study, we used the GGA method for exchange-correlation functionals. For a constant volume, LDA tends to underestimate pressure compared to GGA (Fig. 1). This difference in the predicted pressure is also reflected in the difference

in the viscosity. We note that the LDA-based viscosity tends to be higher than the GGA-based viscosity along an isotherm which can be generally attributed to the over-binding nature of LDA (Supplementary Fig. S5). However, when considered as a function of density, the effects of exchange-correlation functional on melt viscosity are almost negligible (Supplementary Fig. S6). We also find that the effect of composition on the viscosity of silicate melt with similar $SiO_2$ content is rather small (Supplementary Figs. S5, S6).

Our calculated results are in good agreement with the viscosity of MORB melts obtained using the falling sphere method[31] and classical MD simulations[43,67] (Fig. 3, Supplementary Fig. S5). At 2200 K we find that melt viscosity continuously increases with increasing pressure after the viscosity reversal at ~6 GPa. For statistically converged and fully molten simulations at high pressures along all explored isotherms (2200, 3000, and 4000 K), our results indicate that viscosity increases continually with pressure without a second reversal at high pressure as reported in a previous study[46]. Melting and phase relation experiments on MORB indicate that solidification is likely to occur at >30 GPa at ≤2200 K. Even with the experimentally determined high solidus temperatures ($T = 3000$ K[71]), basalt is likely to solidify at $P \geq 60$ GPa. In our study, statistical convergence was not achieved even after 350–400 ps of simulation time for $P \geq -30$ GPa along the 2200 K isotherm (Fig. 2). Similarly, we were unable to obtain statistical convergence for $P \geq -60$ GPa along the 3000 K isotherm (Fig. 2). A previous study documenting the reduction of viscosity at high pressures speculated that an increase in 5, and 7-fold coordinated T-O (T = Al and/or Si) and increasing M-O (M = Ca and/or Mg) coordination with the number of non-bridging oxygens (NBO) as the primary cause for the reduction of viscosity at high pressures[46]. Our results show a continuous decrease of NBOs with increasing pressure along all isotherms (Supplementary Fig. S3).

## Statistical convergence of transport properties

To further explore the effects of statistical convergence on viscosity, we evaluated the melt viscosity at various simulation timescales. We find that using longer simulation timescales yields higher viscosities compared to the shorter simulation timescales (Fig. 3, Supplementary Fig. S7). We also notice that the simulation time has little or no effect on viscosity after a threshold timescale which occurs after statistical convergence is achieved. For example, the calculated viscosity of the basaltic melt at ~0 GPa and 2200 K calculated using 10 ps of simulation time is 0.0066 Pa.s whereas the viscosity calculated using 40 ps of simulation time is 0.031 Pa.s. From the simulation time of 40–120 ps, the viscosity remains constant within an error (Supplementary Fig. S7). Similarly at 33.7 GPa and 2200 K, the viscosity increases by nearly one order of magnitude from 25 to 150 ps and remains unchanged for the rest of the simulation time of 350 ps. However, for non-converged simulations at high-pressure conditions, the melt viscosity continues to increase with increasing simulation time within our simulation timescales and thus requires very long simulation times to become constant (Supplementary Fig. S7). For instance, our viscosity estimated from one ~360 ps run at 47.3 GPa and 2200 K is 5.3 Pa.s. The VFT model predicts the viscosity at 47.3 GPa to be >100 Pa.s (Eq. 2) and thus to have a reliable estimate of the viscosity of melt at this condition, a simulation time >8000 ps (8 ns) is required (Supplementary Fig. S7). An alternate method to estimate the required simulation time is by considering the Maxwell relation for the relaxation time ($\tau_{relax}$) for viscous flow, where $\tau_{relax} = \frac{\eta}{G_\infty}$, with $\eta$ being the viscosity of the melt and $G_\infty$ being the shear modulus at a fully relaxed timescale. It is estimated that at fully relaxed timescales, $G_\infty \approx 10^{10}$ Pa[72]. Thus, $\tau_{relax}$ for the fully relaxed viscous melt with $\eta = 100$ Pa.s requires ~10 ns of simulation time. For an FPMD simulation with more than 200 atoms, it is often unrealistic to simulate for nanoseconds timescale to obtain fully converged results on the transport properties of a viscous melt. Thus, under these conditions, with shorter simulation timescales, the

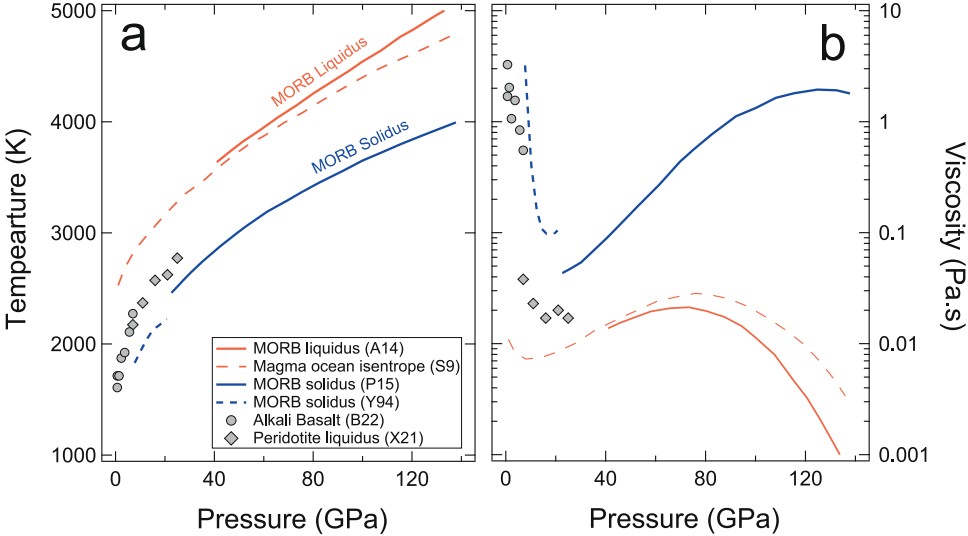

**Fig. 4 | Temperature and viscosity of magma ocean. a** Temperature profiles showing a magma ocean isentrope:S9 (orange dashed lines)[73], liquidus (orange solid lines)[71], and the solidus (solid and dashed blue lines)[78,79] of mid-oceanic ridge basalt (MORB). **b** The predicted viscosity of model basaltic melts using the modified Vogel-Fulcher-Tammann (VFT) model (Eq. 2) along corresponding temperature profiles in panel (**a**). Symbols represent previous experimental data for alkali basalt melt (circles)[34] and peridotite melt (diamonds)[10] along the liquidus.

viscosity results are likely to have large uncertainties and we do not use such results in our analysis and discussion. Moreover, these high-pressure conditions lie within the $P-T$ region of the phase diagram where solid phases are likely to be stable, which makes the melt viscosity at those conditions less meaningful (Fig. 3).

## Implications for crystallization of magma ocean

Along all isotherms, the viscosity of the basaltic melt increases with increasing pressure for most of the mantle pressure regime (Fig. 3). Temperature increases concurrently with pressure along all the temperature profiles of mantle relevance[71,73–75]. During the Archean and Hadean eons, the mantle could have been much hotter following major impacts and radioactive heating[76,77]. To examine the effect of a geothermal gradient on the viscosity of the magma ocean, we extrapolated the viscosity of our basaltic melt to $P-T$ conditions along possible magma ocean isentropes[71,73,74,78–80] using the modified VTF function (Eq. 2). Along any magma ocean temperature profile, the viscosity of the basaltic melt varies non-monotonically with pressure (Fig. 4). The viscosity decreases up to pressures of 8 GPa, which is then followed by an increase in viscosity up to mid-mantle pressure of 70 GPa. Thereafter the viscosity decreases up to the deep mantle pressure of 136 GPa (Fig. 4). The negative pressure dependence of viscosity up to ~8 GPa in our study implies that the magma ocean at a shallower depth is likely to cool faster than previously estimated, assuming an identical thermally conductive lid since the cooling rate of the magma ocean is inversely proportional to its viscosity[8,47,48]. The viscosity ($\eta$) minima at this depth could also have implications in modern Earth as the viscosity of melt has a significant influence on melt mobility ($\Delta\rho/\eta$)[81,82]. Here, the $\Delta\rho$ is the density contrast between the silicate melt and the surrounding mantle. The melt mobility is expected to be high at a depth where the viscosity is minimum. We have extrapolated viscosity to a lower temperature relevant for LAB and we find viscosity at depth is lower than at the surface[31,45]. An upwelling silicate melt with increasing viscosity could thus experience long residence times at a shallower region than the depth of the viscosity minimum. Longer residence times indicate that an upwelling melt is more likely to pond thereby sustaining a partial melt layer, which could explain the seismic anomaly at the lithosphere and asthenosphere boundary (LAB)[31,81,83].

For a completely molten magma ocean at mid-mantle depths, i.e., pressure around 70 GPa, the temperature is around 4000 K for both

the magma ocean adiabat[73] and MORB liquidus[71] (Fig. 4). Based on our results, the viscosity of basaltic magma at such mid-mantle conditions is ~0.027 Pa.s which is smaller than the viscosity of $MgSiO_3$ melt (0.048 Pa.s) at similar conditions[35,40]. The difference in viscosity could be due to the pressure difference induced by the different exchange-correlation functionals used by the previous and present studies. We find the viscosities of the basaltic melt and $MgSiO_3$ melt as a function of density are almost identical (Supplementary Fig. S6). Using the viscosity from our study, we estimate the critical parameters for a dynamical convection model of a magma ocean, including the Rayleigh and Prandtl number given by, $Ra = (\alpha\rho g(T_M - T_S)L^3)/(\kappa\eta)$ and $Pr = (\eta)/(\rho\kappa)$, respectively. Here $T_M$ is the mantle potential temperature, i.e., the lowest temperature at which the mantle would be completely molten, and $T_S$ is a surface temperature set by the dense atmosphere surrounding the magma ocean of early Earth[40]. We estimate $R_a$ and $P_r$ by using $T_M = 2500$ K and $T_S = 1000$ K[40]. For the viscosity ($\eta$), density ($\rho$), and thermal expansivity ($\alpha$), we use the values from this study assuming the depth scale ($L$) of 3000 km. The $g$ and $\kappa$ are acceleration due to gravity and thermal diffusivity, respectively. Thermal diffusivity ($\kappa$) can be estimated using $k/(\rho C_P)$. We adopt the value of thermal conductivity $k = 2.8$ $Wm^{-1}K^{-1}$ at 70 GPa and 4000 K from a recent study[84] and the specific heat ($C_P$) from our simulations. The estimated values of $R_a$ and $P_r$ are $10^{30}$ and 15, respectively, at mid-mantle depth which indicates turbulent convection in the magma ocean. Such convection is likely to significantly influence the crystal settling in the magma ocean[40,85]. The lifetime of the magma ocean is influenced by the viscosity of its constituent silicate melt. The previous estimation for the cooling time of a magma ocean varies significantly, ranging from a few to hundreds of millions of years. Prior geodynamical model using a viscosity of 100 Pa.s estimated that the magma ocean could have survived for ~100–200 Ma[3]. Nevertheless, based on our results, the magma ocean viscosity could be several orders of magnitude lower than 100 Pa.s (Fig. 4). Recently, the timescales of the magma ocean have been revised to a few million years using a more realistic viscosity of 0.1 Pa.s[47,48]. Our results on melt viscosity along potential magma ocean temperature profiles indicate the magma ocean viscosity is ~0.01–0.03 Pa.s for the entire magma ocean, which shows that the cooling times of the magma ocean could be even faster than a few million years (Fig. 4). However, in addition to the influence of viscosity, partial crystallization of the magma ocean and the presence of an insulating atmosphere may also

increase the cooling time of the magma ocean by many orders of magnitude[7,73,86,87]. Crystallization of the magma ocean is predicted to begin in the middle of the mantle where the turbulent flow and viscosity of the magma ocean could alter the settling of crystals[23].

The geochemical signature from the oceanic island basalts, mantle source rocks, and chondritic meteorites often hint toward prevalent heterogeneity in the present-day mantle[88,89]. The crystallization of the magma ocean could have played a significant role in the creation of the inferred geochemical heterogeneity in the mantle[10,30]. If the early Earth was significantly heterogenous, the magma ocean stage should have been followed by a fractional crystallization[49,50]. However, if the equilibrium solidification of the magma ocean is considered, chemical heterogeneity should be added to the mantle much later than the early crystallization of the magma ocean[51]. Thus, the nature of the solidification of the magma ocean can be estimated from the ratio of the grain size of minerals in the magma ocean to the critical grain size[10,30] i.e., the maximum size of grains that can float in the magma ocean. If this ratio is larger than 1, the magma ocean likely followed a fractional crystallization pathway. In contrast, for the grain size ratio smaller than one, the magma ocean solidification likely followed an equilibrium crystallization. The grain size of minerals crystallized from the magma ocean is largely controlled by the viscosity of the magma ocean[7]. A recent geochemical model based on the low viscosity of peridotitic melt that ranges from 0.038 to 0.017 Pa.s and the insulating atmospheric blanketing effect concluded that the solidification of the magma ocean was fractional up to 700 km[10]. Our results on basaltic melt viscosity are similar to that of the peridotitic melt at pressures up to ~25 GPa (Fig. 3). Due to the higher liquidus temperature of basalt compared to that of peridotite, the viscosity of basaltic melt along potential magma ocean temperature is lower than the peridotitic melt and it decreases further below 1000 km (Fig. 4)[10,71,73,74]. The low viscosities of our basaltic melts support a similar hypothesis of fractional crystallization of the magma ocean as a source of mantle heterogeneity[88,89].

## Methods

### First-principles molecular dynamics simulation

First-principles molecular dynamics simulations were employed to study the transport properties of silicate melts with a model basaltic composition. Model basalt is the eutectic composition of 36 mol% anorthite and 64 mol% diopside (with an overall composition of $Ca_{22}Mg_{14}Al_{16}Si_{44}O_{148}$) which has also been studied extensively[38,44,46,54,55,90]. In terms of the weight percentage of oxides, model basaltic melt contains 23.5 wt% CaO, 10.7 wt% MgO, 15.5 wt% $Al_2O_3$, and 50.3 wt% $SiO_2$. We used 244 atoms in simulation supercells. We used generalized gradient approximation (GGA) for the exchange-correlation functional as implemented in the Vienna Ab initio simulation package (VASP)[91–95]. The simulations were set using an *NVT* ensemble, where the number of particles (*N*), the volume (*V*), and the temperature (*T*) of the system remain constant. Simulations were run using the projector augmented wave (PAW) method to compute the interatomic forces[96]. A constant temperature in the simulation was maintained using the Nosé-Hoover thermostat[97]. We set the kinetic energy cut-off of the plane wave to 400 eV and the Brillouin zone sampling to the gamma point. Pulay stress of 2.9–7.2 GPa over the volume range considered in this study was added to the calculated pressures to correct the effect of a finite energy cut-off.

To obtain an equilibrated melt structure, the starting configuration can either be well-equilibrated at higher temperatures or lower densities. After initial equilibration the temperature is lowered[44,59] or the melt structure is gradually compressed to greater densities[98,99]. The high temperatures and lower densities help enhance the mobility of the atoms and thus equilibrate the structure quickly during the simulations. We began our simulations at a high temperature of ≥6000 K for ~100 picoseconds (ps) to obtain a well-equilibrated melt structure. Then the temperature of the melt was decreased at constant

volume (isochore) to desired lower temperatures, i.e., 4000, 3000, and 2200 K. The simulations were performed for ~100–400 ps based on volume-temperature conditions using time steps of 2 femtoseconds (fs). The simulations at high temperatures of 4000 K are highly relevant for the high-temperature conditions of the early Earth magma ocean[8,23]. At lower temperatures, i.e., 2200 K, and high pressures, long simulations up to 400 ps were performed to ensure the required statistical convergence of the results. Typical FPMD simulation times are ~100 ps with the longest reported simulation duration of 240 ps[41,46].

### Effect of simulation cell size on the properties of melt

To evaluate the effect of finite supercell size (244 atoms), we simulated basaltic melt with 122 atoms, 244 atoms, 366 atoms, and 488 atoms, respectively. The result of the cell size test shows that the calculated energy and pressure remain largely unaffected by the system size (Supplementary Fig. S8). We note that the viscosity of the basaltic melt is insensitive to the different cell sizes considered in this study (Supplementary Fig. S9), which is consistent with a recent finding of a negligible size effect on the viscosity of $MgSiO_3$ melt using deep potentials-based molecular dynamics simulations[35]. However, we find that the self-diffusion is sensitive to the size of the simulation cell (Supplementary Fig. S10). We applied the required correction to diffusivity following the scheme proposed in the previous study[35,100] which showed that self-diffusion of water from MD simulation increased as a linear function of $N^{-1/3}$, where *N* represents the number of atoms in the simulation cell (Supplementary Fig. S10).

### Transport properties

We used the blocking method to determine the uncertainty in the pressure from the standard deviation on the pressure fluctuations[101]. At each pressure and temperature condition, we examined the mean square displacement (MSD) and the radial distribution function (RDF) to ensure the melt was in a fully liquid state during the simulation (Fig. 2, Supplementary Fig. S1, Supplementary Data 1). The self-diffusion coefficient of individual species was estimated from the motion of all atoms using the Einstein formulation (Eq. 3)

$$D = \lim_{t \to \infty} \frac{1}{6t} \left( \frac{1}{N} \sum_{i=1}^{N} \{r_i[t + t_0] - r_i[t_0]\}^2 \right) \quad (3)$$

The terms inside the parenthesis refer to the MSD, where $r_i$ and $r_i[t + t_0]$ refer to the position of the *i*th atom at time $t_0$ and its position after a later time *t*.

We used the Green-Kubo relation to estimate the viscosity (η) of silicate melt (Eq. 4)[60].

$$\eta = \frac{V}{10 k_B} \int_0^\infty \left\langle \left( \sum_{\alpha\beta} P_{\alpha\beta}(t_0 + \delta t). P_{\alpha\beta}(t_0) \right) \right\rangle dt \quad (4)$$

where, $P_{\alpha\beta}$ is the symmetrized traceless portion of the stress tensor $\sigma_{\alpha\beta}$ from each simulation step given by $P_{\alpha\beta} = \frac{1}{2}(\sigma_{\alpha\beta} + \sigma_{\beta\alpha}) - \frac{1}{3}\delta_{\alpha\beta}(\sum \sigma_{\lambda\lambda})$. Here, $\delta_{\alpha\beta}$ is the Kronecker delta. We use both the off-diagonal and diagonal components of the stress tensor with weighting factors of 1 and 4/3, respectively, to calculate the melt viscosity (Supplementary Fig. S11)[102,103]. The integrand is a stress autocorrelation function (ACF), which is averaged over time with different origins, $t_0$ for better statistics. Our long simulation durations ensured the ACF was fully converged, i.e., decays to zero before we calculated the melt viscosity.

## Data availability

All the data used in this study are presented in the main paper or in the Supplementary Information and Supplementary Data 1. The processed data used to produce figures are deposited in Supplementary Data 1. The parameters used in the simulation are described in the "Methods" section.

## Code availability

Simulations were performed using the Vienna ab initio simulation package (VASP) software. More details on the simulation package are available at https://www.vasp.at/.

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

## Acknowledgements

M.M., A.A., and S.B. acknowledge the National Science Foundation grants: EAR1763215 and EAR1753125. B.B.K. acknowledges National Science Foundation grant EAR1463807 for funding this research. All authors thank the Extreme Science and Engineering Discovery (XSEDE) supercomputing facilities (TG-GEO170003) and the Research Computing Center (RCC) at Florida State University (FSU) for computing resources.

## Author contributions

S.B., M.M., and B.K. conceived the project. S.B. performed the simulations and wrote first draft of the manuscript. A.A., D.G., M.M., and B.K. contributed to the analysis, discussion, interpretation of the results, writing and revising of the article.

## Competing interests

The authors declare no competing interests.
