## [Peer Review File · Nature Communications]

REVIEWER COMMENTS

Reviewer #1 (Remarks to the Author):

This paper reports viscosity of molten basalt at all pressure conditions relevant of the terrestrial magma ocean. The main finding of the paper is that previously reported viscosity of molten basalt, also from theoretical calculations and published in Nat. Comm. (Majumdar et al. 2020), were wrong, as suspected from the comparison between P-T values and melting relationships. The explanation given is convincing and well documented, thus ending the controversy. Other main conclusions, such as a lower viscosity of the magma ocean than usually considered in cooling models, are valid and very interesting but had already been proposed before, including by the present authors. The implications for fractional vs equilibration crystallisation of magma ocean could be better described and argued here rather than just stated.

To conclude, the present results seem sound and carefully described. Main conclusions are not new, but they do show that results from Majumdar et al. were wrong despite being published in Nat. Comm. I therefore suggest that the present work should also be.

I cannot comment on the calculations. However, please find below some minor points to be corrected:

1- I.51 the authors state that extrapolations of viscosity from low P experiments to high P often add significant uncertainties; this is true but the authors should specify that this is particularly due to the melt changing structure with increased pressure, hence its macroscopic properties such as viscosity also may change.

2- I.71 the melt composition is given in number of atoms. It would be very useful for most readers to have it translated somewhere in at% or wt%.

3- I.74: most readers do not know the simulations time of previous FMPD studies, would be useful to give some numbers here for comparison.

4- I.129: while the authors do compare previous results from Majumdar et al. to published basalt melting relationships, they should do it also for their own results, hence explaining why results obtained at some conditions are to be excluded from the interpretation of their results.

5- I.149: it does not make sense to use terms such as 'network modifiers' and 'network formers' to describe magmas under high pressures as the network of tetrahedral [SiO₄]⁴⁻ units is long gone under pressure; those terms are only valid for ambient (or low P) magmas.

6- I.165/178: could the authors comment on why calculations predict a rise in viscosity from 6 GPa on while experiments report it to 10 GPa? Could this be linked with the overestimation of density by the present method?

Reviewer #2 (Remarks to the Author):

Magma ocean dynamics: insights from transport properties of basaltic melt

The paper present the results of first-principles calculations of the equation of state of basaltic melt at lower mantle conditions. In contrast to a previous study, the current work does not observe a minimum in the viscosity at about 70GPa for isotherms at 2200K and 3000K, instead the viscosity continues to increase with pressure. This difference is attributed to short simulation times and a different method for producing the initial melt structure in the previous study.

The authors then calculate the viscosity using various temperature profiles: magma ocean isentrope, liquidus and solidus of MORB. Based on these the authors suggest that a minimum at about 8GPa, also seen in previous studies, indicates that the magma ocean at shallower depth is likely to cool faster than previously estimated. They also suggest that, due to the relationship between melt viscosity and mobility, as an upwelling reaches these shallower depths, it is more likely to have a long residence time, which could lead to ponding and sustain a partial melt layer. Finally, the authors estimate Rayleigh and Prandtl

numbers and use them to infer that at mid-mantle depths the magma ocean will be in a state of turbulent convection, which would influence settling of crystals. They go on to suggest that their viscosities indicate the magma ocean could have cooled in less than a few millions years, and fractional crystallization could be the origin of current mantle heterogeneity.

From a technical point of view I do not have any criticisms of the method - it is well established. My only concern is that the authors do not mention any test for possible finite-size effects. Either the authors need to show that the simulation cell size has no effect or, at least, provide an indication of the expected direction and magnitude in which their results would change if a larger cell were used.

In regard to the significance of the work, I am unfamiliar with the literature on silicate melts. The viscosity decrease at low pressures appears to have been reported previously. Other implications are speculative, but sensible and logical, and what you would expect for this kind of investigation. What is clear is that the high-pressure results of the present work corrects serious errors in and extends a study of basaltic silicate melt recently published in Nature Communications (Majumdar et al. 2020) and, as such, merits publication in the same journal.

Some minor comments are listed below:

L110 I would make it clear that they are discussing K0. The bulk moduli of mantle minerals are greater than a 100GPa at lower mantle conditions.

L127-129 What is 'atom-atom bond lengths', do you just mean 'bond lengths'?

L131 Why did the authors not simply run the unconverged simulations longer? Looking at Figure 3, given that Majumdar et al. (2020) ran their simulations for between 120 and 240 ps, it seems that simulation time cannot be the main reason for the difference between the two sets of results and that the method of producing the melt structure and P-T conditions is important. It might have been useful for the authors to try to reproduce the results of Majumdar et al. (2020) at one P-T point to confirm this.

L138-139 The authors state that 'When the ACF decays to zero within the time scale of a simulation, it indicates that the Maxwell relaxation time of basaltic melts are shorter than seismic periods.' I think I know what the authors are trying to say here, but please consider re-phrasing, to make it clearer. Of course, the truth of this statement depends on the length of the simulation.

L299 What low viscosity value did the authors use? Please state.

L310 Unclear how references 85 and 86 relate to simulation cells.

Figure 1 I would replace the label 'not used' with 'solid phase'.

The text still needs some work. There are quite a few errors or typos. I have suggested some change below that may help the authors to tidy up the text.

L32 What is meant by 'overall mantle'?

L41 Consider replacing 'that produce' with 'that produces'.

L62 Consider replacing 'show the' with 'show'.

L65 Consider replacing 'the observed' with 'observed'.

L70 Consider replacing 'the present-day' with 'present day'.

L73 Consider replacing 'at high temperature of' with 'at a high temperature of'.

L78 Consider removing 'serves as an analog composition for the magma ocean'.

L81 Consider replacing 'could have been' with 'could have'.

L92 Consider replacing 'large increment with compression' with 'strong dependence on compression'.

L93 Consider replacing 'i.e. an order of magnitude changes' with 'it changes by an order of magnitude'.

L97 Consider replacing 'At all' with 'Along all'.

L98 Consider replacing 'with GGA method' with 'with the GGA'.

L96 Consider replacing 'solid regime' with 'the solid regime'.

L108 Consider replacing 'previous study' with 'the previous study'.

L109 Consider replacing 'mantle mineral' with 'mantle minerals'.

L112 Consider replacing 'atomistic scale structural' with 'atomic scale structure'.

L237 Consider replacing 'method contrasts the' with 'method contrasts with the'.

L281 Consider replacing 'from previous study' with 'from a previous study'.

L286 Consider replacing 'of magma' with 'of a magma'.

L286 Consider replacing 'of magma ocean' with 'of a magma ocean'.

L319 Consider replacing 'at high' with 'at a high'.

L319 Consider replacing 'obtain the' with 'obtain a'.

L320 Consider replacing 'constant volumes' with 'constant volume'.

L331 Consider replacing 'additional time' with 'later time'.

L598 Consider replacing 'melt form' with 'melt from'.

L614 Unclear why (a and b) is in bold font.

Figure 4 Consider replacing 'isentrope' with 'isentropes'

Supplementary material

L17 Consider replacing 'is given by' with 'from'.

Reviewer #3 (Remarks to the Author):

The authors explore the viscosity of silicate liquid at conditions of the magma ocean that may have occupied much or all of the mantle of the early Earth. The authors find, on the basis of first principles molecular dynamic simulations that the viscosity of the magma ocean is very low (<0.1 Pa s) and that it increases monotonically with increasing pressure at high pressure. The latter result is at odds with another recent study, and the present authors do a nice job of highlighting the shortcomings of the previous study. In particular, the present authors use much longer run times and a much more careful analysis of ergodicity to demonstrate the accuracy of this results.

I recommend publication after consideration of the following points:

1) I. 116. "Limited available space". This is ambiguous and I don't think its what the authors mean. I think they mean "diminished free volume" or something like it.

2) II. 260-267. I do not completely follow the arguments here. They start in the context of the magma ocean, and end at the LAB. The problem is that the LAB is much colder than the temperature being considered. Are the authors extrapolating their results to the conditions of the LAB in order to make this argument?

3) I. 271. Why does basaltic melt have a lower viscosity than MgSiO_3 melt? Can some rationalization or insight be provided?

4) I. 281. The authors should use instead values of the thermal conductivity of a silicate melt determined at high pressure for the first time via ab initio simulation (GRL, 2021).

5) Last paragraph. The authors relate their viscosity to the cooling time of the magma ocean via the Rayleigh number and scaling laws from the literature. This is fine as far as it goes, but should at least mention some other very important, probably controlling factors: the presence of an insulating atmosphere, and the effect of partial crystallization. Either of these factors can change the cooling time by many orders of magnitude and are probably more important considerations than a few orders of magnitude difference in melt viscosity.

6) II. 296-299. This sentence does not make sense. It could be abbreviated: "If significant chemical heterogeneity is already present...then chemical heterogeneity is present."

7) Last paragraph. The arguments for fractional crystallization is not clearly explained. Why does low viscosity lead to fractional crystallization exactly? Large melt mobility? Large crystal settling velocity? The authors need to clarify.

8) I. 335. The authors should justify their choice of 4/3 weighting. They could show the stress auto-correlation function and the viscosity computed from each individual stress component.

REVIEWER COMMENTS

This file contains the rebuttal letter with point-by-point reply to reviewer's comments. The major changes in the manuscript are highlighted in yellow.

Reviewer #1 (Remarks to the Author):

This paper reports viscosity of molten basalt at all pressure conditions relevant of the terrestrial magma ocean. The main finding of the paper is that previously reported viscosity of molten basalt, also from theoretical calculations and published in Nat. Comm. (Majumdar et al. 2020), were wrong, as suspected from the comparison between P-T values and melting relationships. The explanation given is convincing and well documented, thus ending the controversy. Other main conclusions, such as a lower viscosity of the magma ocean than usually considered in cooling models, are valid and very interesting but had already been proposed before, including by the present authors. The implications for fractional vs equilibration crystallization of magma ocean could be better described and argued here rather than just stated.

To conclude, the present results seem sound and carefully described. Main conclusions are not new, but they do show that results from Majumdar et al. were wrong despite being published in Nat. Comm. I therefore suggest that the present work should also be.

The authors would like to thank the reviewer for the suggestion.

I cannot comment on the calculations. However, please find below some minor points to be corrected:

1- l.51 the authors state that extrapolations of viscosity from low P experiments to high P often add significant uncertainties; this is true but the authors should specify that this is particularly due to the melt changing structure with increased pressure, hence its macroscopic properties such as viscosity also may change.

Thank you for your suggestions. We have modified the manuscript to address this issue (lines # 47-50).

2- l.71 the melt composition is given in number of atoms. It would be very useful for most readers to have it translated somewhere in at% or wt%.

We have provided mol% and wt.% in the revised manuscript to address this suggestion (lines # 74-76).

3- l.74: most readers do not know the simulations time of previous FMPD studies, would be useful to give some numbers here for comparison.

Thank you for your suggestions. We have modified the manuscript to clarify the issue (lines # 82-83)

4- l.129: while the authors do compare previous results from Majumdar et al. to published basalt melting relationships, they should do it also for their own results, hence explaining why results obtained at some conditions are to be excluded from the interpretation of their results.

Thank you for your suggestions. In the revised manuscript, we have compared the MSD with our previous work (lines # 135-137)

5- l.149: it does not make sense to use terms such as 'network modifiers' and 'network formers' to describe magmas under high pressures as the network of tetrahedral [SiO₄]⁴⁻ units is long gone under pressure; those terms are only valid for ambient (or low P) magmas.

We agree with the reviewer's comment that the terms 'network modifiers' and 'network formers' are only valid at ambient pressures. In the revised manuscript, we have modified the manuscript to avoid these terms throughout the paper and replaced these words with the names of the atom/species.

6- l.165/178: could the authors comment on why calculations predict a rise in viscosity from 6 GPa on while experiments report it to 10 GPa? Could this be linked with the overestimation of density by the present method?

Thank you for raising this issue. We feel that the prior version of the draft might have contributed to the confusion. We made a qualitative comparison with both the previous computational and experimental studies, and hence we were referring to below 10 GPa. The plausible reasons for the discrepancies could be related to the exact melt composition and temperature conditions. We find the effect of the method is small on the pressure condition of viscosity minima. In the revised manuscript, we have now modified the sentence to avoid such confusion (lines # 194-201).

Reviewer #2 (Remarks to the Author):

The paper present the results of first-principles calculations of the equation of state of basaltic melt at lower mantle conditions. In contrast to a previous study, the current work does not observe a minimum in the viscosity at about 70GPa for isotherms at 2200K and 3000K, instead the viscosity continues to increase with pressure. This difference is attributed to short simulation times and a different method for producing the initial melt structure in the previous study.

The authors then calculate the viscosity using various temperature profiles: magma ocean isentrope, liquidus and solidus of MORB. Based on these the authors suggest that a minimum at about 8GPa, also seen in previous studies, indicates that the magma ocean at shallower depth is likely to cool faster than previously estimated. They also suggest that, due to the relationship between melt viscosity and mobility, as an upwelling reaches these shallower depths, it is more likely to have a long residence time, which could lead to ponding and sustain a partial melt layer. Finally, the authors estimate Rayleigh and Prandtl numbers and use them to infer that at mid-mantle depths the magma ocean will be in a state of turbulent convection, which would influence settling of crystals. They go on to suggest that their viscosities indicate the magma ocean could have cooled in less than a few millions years, and fractional crystallization could be the origin of current mantle heterogeneity.

From a technical point of view I do not have any criticisms of the method - it is well established.

My only concern is that the authors do not mention any test for possible finite-size effects. Either the authors need to show that the simulation cell size has no effect or, at least, provide an indication of the expected direction and magnitude in which their results would change if a larger cell were used.

The authors would like to thank the reviewer for the helpful suggestions. To explore the effect of finite supercell size, we have simulated basaltic melt with system sizes 122, 244 (used in this study), 366, and 488 atoms, respectively. We have added a discussion of possible finite-size effects on the revised version of the manuscript. Please refer to lines #353-362 of the revised manuscript (**Supplementary Figures S8-S10**).

In regard to the significance of the work, I am unfamiliar with the literature on silicate melts. The viscosity decrease at low pressures appears to have been reported previously. Other implications are speculative, but sensible and logical, and what you would expect for this kind of investigation. What is clear is that the high-pressure results of the present work corrects serious errors in and extends a study of basaltic silicate melt recently published in Nature Communications (Majumdar et al. 2020) and, as such, merits publication in the same journal.

Thank you for the comments and recommendation.
Some minor comments are listed below:

L110 I would make it clear that they are discussing K0. The bulk moduli of mantle minerals are greater than a 100GPa at lower mantle conditions.

Thank you for pointing it out. We have modified the sentence to make it clear (lines #110-112).

L127-129 What is 'atom-atom bond lengths', do you just mean 'bond lengths' ?

Yes. Thank you for pointing it out. We have corrected this in the revised manuscript.

L131 Why did the authors not simply run the unconverged simulations longer?

Looking at Figure 3, given that Majumdar et al. (2020) ran their simulations for between 120 and 240 ps, it seems that simulation time cannot be the main reason for the difference between the two sets of results and that the method of producing the melt structure and P-T conditions is important. It might have been useful for the authors to try to reproduce the results of Majumdar et al. (2020) at one P-T point to confirm this.

Thank you for raising this issue. We agree with the reviewer that the simulation time is not the only reason for the difference between the studies. Because the atoms diffuse very slowly, the simulation may not converge by simply running for a longer time at most of the pressure and temperature conditions. To illustrate, our viscosity results for one of the unconverged simulations (~360 ps) at 47.3 GPa and 2200 K is 5.3 Pa.s. The VFT model predicts the viscosity at 47.3 GPa to be > 100 Pa.s (equation 2 in the manuscript) and thus it we require a timescale of >8000 picoseconds (8 ns) to get that value (**Supplementary Figure S7**). Running simulation for that long is simply unrealistic for FPMD simulation with 244 atoms. Another way to estimate the required simulation time is using the Maxwell relation of relaxation time (τ_{relax}) for viscous flow $\tau_{relax} = \frac{\eta}{G_{\infty}}$, where η is melt viscosity and G_{∞} is shear modulus at infinite frequency.

Typical value of $G_{\infty} \approx 10^{10}$ Pascals¹. Thus, the τ_{relax} for the fully relaxed viscous melt with

$\eta=100$ Pa.s requires ~ 10 ns of simulation time. So, it is not realistic for FPMD simulation with >200 atoms to simply run longer time in order to get the converged result for a fully relaxed melt. Moreover, basalt is in the solid phase of the phase diagram at pressure ≥ 34 GPa and 2200 K in all prior studies (**Figure 1**). In some of the prior studies, basalt is outside of the solidus line at $P > 18$ GPa and 2200 K^{2,3}. Therefore, the results will be less useful even if we somehow get the required convergence by running simulations for a very long time. We have discussed this in the revised manuscript (lines # 154-162).

Although it would be very useful for us to try to reproduce the results of Majumdar et al. (2020)⁴ at one point using an unequilibrated melt structure, it may not be straightforward to do so. The results from MD simulation from unequilibrated melt structure could be uncertain and highly depend on the initial configuration because the melt was never equilibrated at high temperatures. So, unfortunately, it is not possible for us to reproduce the results of Majumdar et al. (2020).

L138-139 The authors state that ‘When the ACF decays to zero within the time scale of a simulation, it indicates that the Maxwell relaxation time of basaltic melts are shorter than seismic periods.’ I think I know what the authors are trying to say here, but please consider re-phrasing, to make it clearer. Of course, the truth of this statement depends on the length of the simulation.

This is a great point. We have rephrased the sentence to make it clear (lines # 143-146).

L299 What low viscosity value did the authors use? Please state.

We have revised the manuscript to address this issue (line # 320).

L310 Unclear how references 85 and 86 relate to simulation cells.

Thank you for pointing this out. We were referring to the *first principles* molecular dynamics. We have now moved the references to the correct location (line # 333-334).

Figure 1 I would replace the label ‘not used’ with ‘solid phase’.

This is a great suggestion. We have modified the figure. Please refer to Figure 1 of the revised draft.

The text still needs some work. There are quite a few errors or typos. I have suggested some change below that may help the authors to tidy up the text.

Thank you for taking the time to point out errors and/or typos. We have thoroughly revised the manuscript to minimize errors or typo.

L32 What is meant by ‘overall mantle’?

We mean from upper mantle to deep lower mantle. We have modified the sentence to avoid confusion.

L41 Consider replacing ‘that produce’ with ‘that produces’.

We have fixed this error in the revised manuscript.

L62 Consider replacing ‘show the’ with ‘show’.

We have fixed this error in the revised manuscript.

L65 Consider replacing ‘the observed’ with ‘observed’.

We have fixed this error in the revised manuscript.

L70 Consider replacing 'the present-day' with 'present day'.

We have fixed this error in the revised manuscript.

L73 Consider replacing 'at high temperature of' with 'at a high temperature of'.

We have fixed this error in the revised manuscript.

L78 Consider removing 'serves as an analog composition for the magma ocean'.

We have fixed this error in the revised manuscript.

L81 Consider replacing 'could have been' with 'could have'.

We have fixed this error in the revised manuscript.

L92 Consider replacing 'large increment with compression' with 'strong dependence on compression'.

We have fixed this error in the revised manuscript.

L93 Consider replacing 'i.e. an order of magnitude changes' with 'it changes by an order of magnitude'.

We have fixed this error in the revised manuscript.

L97 Consider replacing 'At all' with 'Along all'.

We have fixed this error in the revised manuscript.

L98 Consider replacing 'with GGA method' with 'with the GGA'.

We have fixed this error in the revised manuscript.

L96 Consider replacing 'solid regime' with 'the solid regime'.

We have fixed this error in the revised manuscript.

L108 Consider replacing 'previous study' with 'the previous study'.

We have fixed this error in the revised manuscript.

L109 Consider replacing 'mantle mineral' with 'mantle minerals'.

We have fixed this error in the revised manuscript.

L112 Consider replacing 'atomistic scale structural' with 'atomic scale structure'.

We have fixed this error in the revised manuscript.

L237 Consider replacing 'method contrasts the' with 'method contrasts with the'.

We have fixed this error in the revised manuscript.

L281 Consider replacing 'from previous study' with 'from a previous study'.

We have fixed this error in the revised manuscript.

L286 Consider replacing 'of magma' with 'of a magma'.

We have fixed this error in the revised manuscript.

L286 Consider replacing 'of magma ocean' with 'of a magma ocean'.

We have fixed this error in the revised manuscript.

L319 Consider replacing 'at high' with 'at a high'.

We have fixed this error in the revised manuscript.

L319 Consider replacing 'obtain the' with 'obtain a'.

We have fixed this error in the revised manuscript.

L320 Consider replacing 'constant volumes' with 'constant volume'.

We have fixed this error in the revised manuscript.

L331 Consider replacing 'additional time' with 'later time'.

We have fixed this error in the revised manuscript.

L598 Consider replacing 'melt form' with 'melt from'.

We have fixed this error in the revised manuscript.

L614 Unclear why (a and b) is in bold font.

We have fixed this error in the revised manuscript.

Figure 4 Consider replacing 'isentorpe' with 'isentrope'

We have fixed this error in the revised manuscript.

Supplementary material

L17 Consider replacing 'is given by' with 'from'.

Thank you for pointing out. We have fixed this in the revised supplementary document.

Reviewer #3 (Remarks to the Author):

The authors explore the viscosity of silicate liquid at conditions of the magma ocean that may have occupied much or all of the mantle of the early Earth. The authors find, on the basis of first principles molecular dynamic simulations that the viscosity of the magma ocean is very low ($<0.1 \text{ Pa s}$) and that it increases monotonically with increasing pressure at high pressure. The latter result is at odds with another recent study, and the present authors do a nice job of highlighting the shortcomings of the previous study. In particular, the present authors use much longer run times and a much more careful analysis of ergodicity to demonstrate the accuracy of this results.

I recommend publication after consideration of the following points:

The authors thank the reviewer for the recommendation.

1) l. 116. "Limited available space". This is ambiguous and I don't think its what the authors mean. I think they mean "diminished free volume" or something like it.

We have modified the sentence to correct it (lines # 116-118). Thank you for your suggestion.

2) ll. 260-267. I do not completely follow the arguments here. They start in the context of the magma ocean, and end at the LAB. The problem is that the LAB is much colder than the temperature being considered. Are the authors extrapolating their results to the conditions of the LAB in order to make this argument?

Thank you for raising this important question. Our simulation at 2200 K is not much hotter compared to the potential temperature at LAB. We have extrapolated viscosity to lower temperature relevant for LAB^{5,6}. However, our interpretation is mostly qualitative based on the viscosity minima at ~4 GPa which is roughly 120 km depth. Since viscosity is lowest at a depth below 90 km, magma mobility is highest below LAB. For the present Earth's interior, it implies that upwelling magma tends to cumulate at a shallower depth of LAB due to increased mobility as suggested in previous experiment⁵. We have revised the draft to address this issue (lines # 266-268).

3) l. 271. Why does basaltic melt have a lower viscosity than MgSiO₃ melt? Can some rationalization or insight be provided?

Thank you for your suggestion. Some of the discrepancies between previous FPMD simulation of MgSiO₃ and basaltic melt can be attributed to difference in exchange correlation functional (LDA vs GGA method). Melt composition have small effect on viscosity when compared as a function of density (**Supplementary Figure S6**). We added more discussion in the revised manuscript (lines # 212-215). We have also added viscosity of MgSiO₃ melt in **Supplementary Figure S5-S6** to demonstrate this point.

4) l. 281. The authors should use instead values of the thermal conductivity of a silicate melt determined at high pressure for the first time via ab initio simulation (GRL, 2021).

Thank you for your suggestion. We have revised the values of thermal conductivity using recent study of thermal conductivity of silicate melt at high pressure and 4000 K⁷ and also revised Rayleigh and Prandtl number based on new thermal conductivity data (lines # 288-289).

5) Last paragraph. The authors relate their viscosity to the cooling time of the magma ocean via the Rayleigh number and scaling laws from the literature. This is fine as far as it goes, but should at least mention some other very important, probably controlling factors: the presence of an insulating atmosphere, and the effect of partial crystallization. Either of these factors can change the cooling time by many orders of magnitude and are probably more important considerations than a few orders of magnitude difference in melt viscosity.

Thank you for your suggestion. We have modified the discussion to address this issue (lines # 301-306).

6) ll. 296-299. This sentence does not make sense. It could be abbreviated: "If significant chemical heterogeneity is already present...then chemical heterogeneity is present."

Thank you for pointing it out. We have modified this portion of manuscript.

7) Last paragraph. The arguments for fractional crystallization is not clearly explained. Why does low viscosity lead to fractional crystallization exactly? Large melt mobility? Large crystal settling velocity? The authors need to clarify.

We rewrote this paragraph to explain the arguments more clearly. Please refer to lines #307-330 of the revised manuscript. Thank you for your suggestions.

8) l. 335. The authors should justify their choice of 4/3 weighting. They could show the stress auto-correlation function and the viscosity computed from each individual stress component.

Thank you for the suggestion. We have added a supplementary figure to show the stress auto-correlation function and the viscosity of individual stress components (**Supplementary Figure S11**).

References

- 1 Dingwell, D. B. & Webb, S. L. Structural relaxation in silicate melts and non-Newtonian melt rheology in geologic processes. *Physics and Chemistry of Minerals* **16**, 508-516 (1989).
- 2 Yasuda, A., Fujii, T. & Kurita, K. Melting phase relations of an anhydrous mid-ocean ridge basalt from 3 to 20 GPa: Implications for the behavior of subducted oceanic crust in the mantle. *Journal of Geophysical Research: Solid Earth* **99**, 9401-9414 (1994).
- 3 Hirose, K., Fei, Y., Ma, Y. & Mao, H.-K. The fate of subducted basaltic crust in the Earth's lower mantle. *Nature* **397**, 53-56 (1999).
- 4 Majumdar, A., Wu, M., Pan, Y., Itaka, T. & John, S. T. Structural dynamics of basaltic melt at mantle conditions with implications for magma oceans and superplumes. *Nature communications* **11**, 1-9 (2020).
- 5 Sakamaki, T. *et al.* Ponded melt at the boundary between the lithosphere and asthenosphere. *Nat. Geosci.* **6**, 1041-1044 (2013).
- 6 Bajgain, S. K. & Mookherjee, M. Carbon bearing aluminosilicate melt at high pressure. *Geochimica et Cosmochimica Acta* **312**, 106-123, doi:10.1016/j.gca.2021.07.039 (2021).
- 7 Deng, J. & Stixrude, L. Thermal conductivity of silicate liquid determined by machine learning potentials. *Geophysical Research Letters* **48**, e2021GL093806, doi:doi.org/10.1029/2021GL093806 (2021).

REVIEWERS' COMMENTS

Reviewer #1 (Remarks to the Author):

The authors have addressed the questions raised. I recommend publication of the manuscript.

Reviewer #2 (Remarks to the Author):

The authors have provided thorough responses to the comments and queries in my original review, in particular, to comment on finite-size effect and I suggest that it is now ready to be published.

This file contains the rebuttal letter with point-by-point replies to the reviewer's comments. Reviewer's comments are in green, and the author's response is indicated by black color.

REVIEWERS' COMMENTS ROUND 1

Reviewer #1 (Remarks to the Author):

This paper reports viscosity of molten basalt at all pressure conditions relevant of the terrestrial magma ocean. The main finding of the paper is that previously reported viscosity of molten basalt, also from theoretical calculations and published in Nat. Comm. (Majumdar et al. 2020), were wrong, as suspected from the comparison between P-T values and melting relationships. The explanation given is convincing and well documented, thus ending the controversy. Other main conclusions, such as a lower viscosity of the magma ocean than usually considered in cooling models, are valid and very interesting but had already been proposed before, including by the present authors. The implications for fractional vs equilibration crystallization of magma ocean could be better described and argued here rather than just stated. To conclude, the present results seem sound and carefully described. Main conclusions are not new, but they do show that results from Majumdar et al. were wrong despite being published in Nat. Comm. I therefore suggest that the present work should also be.

The authors would like to thank the reviewer for the suggestion.

I cannot comment on the calculations. However, please find below some minor points to be corrected:

1- 1.51 the authors state that extrapolations of viscosity from low P experiments to high P often add significant uncertainties; this is true but the authors should specify that this is particularly due to the melt changing structure with increased pressure, hence its macroscopic properties such as viscosity also may change.

Thank you for your suggestions. We have modified the manuscript to address this issue (lines # 47-50).

2- 1.71 the melt composition is given in number of atoms. It would be very useful for most readers to have it translated somewhere in at% or wt%.

We have provided mol% and wt.% in the revised manuscript to address this suggestion (lines # 74-76).

3- 1.74: most readers do not know the simulations time of previous FMPD studies, would be useful to give some numbers here for comparison.

Thank you for your suggestions. We have modified the manuscript to clarify the issue (lines # 82-83)

4- 1.129: while the authors do compare previous results from Majumdar et al. to published basalt melting relationships, they should do it also for their own results, hence explaining why results obtained at some conditions are to be excluded from the interpretation of their results.

Thank you for your suggestions. In the revised manuscript, we have compared the MSD with our previous work (lines # 135-137)

5- l.149: it does not make sense to use terms such as 'network modifiers' and 'network formers' to describe magmas under high pressures as the network of tetrahedral [SiO₄]⁴⁻ units is long gone under pressure; those terms are only valid for ambient (or low P) magmas.

We agree with the reviewer's comment that the terms 'network modifiers' and 'network formers' are only valid at ambient pressures. In the revised manuscript, we modified the manuscript to avoid these terms throughout the paper and replaced these words with the names of the atom/species.

6- l.165/178: could the authors comment on why calculations predict a rise in viscosity from 6 GPa on while experiments report it to 10 GPa? Could this be linked with the overestimation of density by the present method?

Thank you for raising this issue. We feel that the prior version of the draft might have contributed to the confusion. We made a qualitative comparison with both the previous computational and experimental studies, and hence we were referring to below 10 GPa. The plausible reasons for the discrepancies could be related to the exact melt composition and temperature conditions. We find the effect of the method is small on the pressure condition of viscosity minima. In the revised manuscript, we have now modified the sentence to avoid such confusion (lines # 194-201).

Reviewer #2 (Remarks to the Author):

The paper present the results of first-principles calculations of the equation of state of basaltic melt at lower mantle conditions. In contrast to a previous study, the current work does not observe a minimum in the viscosity at about 70GPa for isotherms at 2200K and 3000K, instead the viscosity continues to increase with pressure. This difference is attributed to short simulation times and a different method for producing the initial melt structure in the previous study.

The authors then calculate the viscosity using various temperature profiles: magma ocean isentrope, liquidus and solidus of MORB. Based on these the authors suggest that a minimum at about 8GPa, also seen in previous studies, indicates that the magma ocean at shallower depth is likely to cool faster than previously estimated. They also suggest that, due to the relationship between melt viscosity and mobility, as an upwelling reaches these shallower depths, it is more likely to have a long residence time, which could lead to ponding and sustain a partial melt layer. Finally, the authors estimate Rayleigh and Prandtl numbers and use them to infer that at mid-mantle depths the magma ocean will be in a state of turbulent convection, which would influence settling of crystals. They go on to suggest that their viscosities indicate the magma ocean could have cooled in less than a few millions years, and fractional crystallization could be the origin of current mantle heterogeneity.

From a technical point of view I do not have any criticisms of the method - it is well established.

My only concern is that the authors do not mention any test for possible finite-size effects. Either the authors need to show that the simulation cell size has no effect or, at least, provide an indication of the expected direction and magnitude in which their results would change if a larger cell were used.

The authors would like to thank the reviewer for the helpful suggestions. To explore the effect of finite supercell size, we have simulated basaltic melt with system sizes 122, 244 (used in this study), 366, and 488 atoms, respectively. We have added a discussion of possible finite-size effects on the revised version of the manuscript. Please refer to lines #353-362 of the revised manuscript (**Supplementary Figures S8-S10**).

In regard to the significance of the work, I am unfamiliar with the literature on silicate melts. The viscosity decrease at low pressures appears to have been reported previously. Other implications are speculative, but sensible and logical, and what you would expect for this kind of investigation. What is clear is that the high-pressure results of the present work corrects serious errors in and extends a study of basaltic silicate melt recently published in Nature Communications (Majumdar et al. 2020) and, as such, merits publication in the same journal.

Thank you for the comments and recommendations.

Some minor comments are listed below:

L110 I would make it clear that they are discussing K0. The bulk moduli of mantle minerals are greater than a 100GPa at lower mantle conditions.

Thank you for pointing it out. We have modified the sentence to make it clear (lines #110-112).

L127-129 What is 'atom-atom bond lengths', do you just mean 'bond lengths' ?

Yes. Thank you for pointing it out. We have corrected this in the revised manuscript.

L131 Why did the authors not simply run the unconverged simulations longer?

Looking at Figure 3, given that Majumdar et al. (2020) ran their simulations for between 120 and 240 ps, it seems that simulation time cannot be the main reason for the difference between the two sets of results and that the method of producing the melt structure and P-T conditions is important. It might have been useful for the authors to try to reproduce the results of Majumdar et al. (2020) at one P-T point to confirm this.

Thank you for raising this issue. We agree with the reviewer that the simulation time is not the only reason for the difference between the studies. Because the atoms diffuse very slowly, the simulation may not converge by simply running for a longer time at most of the pressure and temperature conditions. To illustrate, our viscosity results for one of the unconverged simulations (~360 ps) at 47.3 GPa and 2200 K is 5.3 Pa.s. The VFT model predicts the viscosity at 47.3 GPa to be > 100 Pa.s (equation 2 in the manuscript) and thus it requires a timescale of > 8000 picoseconds (8 ns) to get that value (**Supplementary Figure S7**). Simulating for that long is simply unrealistic for FPMD simulation with 244 atoms. Another way to estimate the required simulation time is using the Maxwell relation of relaxation time (τ_{relax}) for viscous flow $\tau_{relax} = \frac{\eta}{G_{\infty}}$, where η is melt viscosity and G_{∞} is shear modulus at infinite frequency. The typical value of $G_{\infty} \approx 10^{10}$ Pascals¹. Thus, the τ_{relax} for the fully relaxed viscous melt with

$\eta=100$ Pa.s requires ~ 10 ns of simulation time. So, it is not realistic for FPMD simulation with >200 atoms to simply run a longer time to get the converged result for a fully relaxed melt. Moreover, basalt is in the solid phase of the phase diagram at pressure ≥ 34 GPa and 2200 K in all prior studies (**Figure 1**). In some of the prior studies, basalt is outside of the solidus line at $P > 18$ GPa and 2200 K^{2,3}. Therefore, the results will be less useful even if we somehow get the required convergence by running simulations for a very long time. We have discussed this in the revised manuscript (lines # 154-162).

Although it would be very useful for us to try to reproduce the results of Majumdar et al. (2020)⁴ at one point using an unequilibrated melt structure, it may not be straightforward to do so. The results from MD simulation from unequilibrated melt structure could be uncertain and highly dependent on the initial configuration because the melt was never equilibrated at high temperatures. So, unfortunately, it is not possible for us to reproduce the results of Majumdar et al. (2020).

L138-139 The authors state that ‘When the ACF decays to zero within the time scale of a simulation, it indicates that the Maxwell relaxation time of basaltic melts are shorter than seismic periods.’ I think I know what the authors are trying to say here, but please consider re-phrasing, to make it clearer. Of course, the truth of this statement depends on the length of the simulation.

This is a great point. We have rephrased the sentence to make it clear (lines # 143-146).

L299 What low viscosity value did the authors use? Please state.

We have revised the manuscript to address this issue (line # 320).

L310 Unclear how references 85 and 86 relate to simulation cells.

Thank you for pointing this out. We were referring to the *first principles* molecular dynamics. We have now moved the references to the correct location (line # 333-334).

Figure 1 I would replace the label ‘not used’ with ‘solid phase’.

This is a great suggestion. We have modified the figure. Please refer to Figure 1 of the revised draft.

The text still needs some work. There are quite a few errors or typos. I have suggested some change below that may help the authors to tidy up the text.

Thank you for taking the time to point out errors and/or typos. We have thoroughly revised the manuscript to minimize errors or typos.

L32 What is meant by ‘overall mantle’?

We mean from the upper mantle to the deep lower mantle. We have modified the sentence to avoid confusion.

L41 Consider replacing ‘that produce’ with ‘that produces’.

We have fixed this error in the revised manuscript.

L62 Consider replacing ‘show the’ with ‘show’.

We have fixed this error in the revised manuscript.

L65 Consider replacing 'the observed' with 'observed'.

We have fixed this error in the revised manuscript.

L70 Consider replacing 'the present-day' with 'present day'.

We have fixed this error in the revised manuscript.

L73 Consider replacing 'at high temperature of' with 'at a high temperature of'.

We have fixed this error in the revised manuscript.

L78 Consider removing 'serves as an analog composition for the magma ocean'.

We have fixed this error in the revised manuscript.

L81 Consider replacing 'could have been' with 'could have'.

We have fixed this error in the revised manuscript.

L92 Consider replacing 'large increment with compression' with 'strong dependence on compression'.

We have fixed this error in the revised manuscript.

L93 Consider replacing 'i.e. an order of magnitude changes' with 'it changes by an order of magnitude'.

We have fixed this error in the revised manuscript.

L97 Consider replacing 'At all' with 'Along all'.

We have fixed this error in the revised manuscript.

L98 Consider replacing 'with GGA method' with 'with the GGA'.

We have fixed this error in the revised manuscript.

L96 Consider replacing 'solid regime' with 'the solid regime'.

We have fixed this error in the revised manuscript.

L108 Consider replacing 'previous study' with 'the previous study'.

We have fixed this error in the revised manuscript.

L109 Consider replacing 'mantle mineral' with 'mantle minerals'.

We have fixed this error in the revised manuscript.

L112 Consider replacing 'atomistic scale structural' with 'atomic scale structure'.

We have fixed this error in the revised manuscript.

L237 Consider replacing 'method contrasts the' with 'method contrasts with the'.

We have fixed this error in the revised manuscript.

L281 Consider replacing 'from previous study' with 'from a previous study'.

We have fixed this error in the revised manuscript.

L286 Consider replacing 'of magma' with 'of a magma'.

We have fixed this error in the revised manuscript.

L286 Consider replacing 'of magma ocean' with 'of a magma ocean'.

We have fixed this error in the revised manuscript.

L319 Consider replacing 'at high' with 'at a high'.

We have fixed this error in the revised manuscript.

L319 Consider replacing 'obtain the' with 'obtain a'.

We have fixed this error in the revised manuscript.

L320 Consider replacing 'constant volumes' with 'constant volume'.

We have fixed this error in the revised manuscript.

L331 Consider replacing 'additional time' with 'later time'.

We have fixed this error in the revised manuscript.

L598 Consider replacing 'melt form' with 'melt from'.

We have fixed this error in the revised manuscript.

L614 Unclear why (a and b) is in bold font.

We have fixed this error in the revised manuscript.

Figure 4 Consider replacing 'isentorpe' with 'isentropo'.

We have fixed this error in the revised manuscript.

Supplementary material

L17 Consider replacing 'is given by' with 'from'.

Thank you for pointing that out. We have fixed this in the revised supplementary document.

Reviewer #3 (Remarks to the Author):

The authors explore the viscosity of silicate liquid at conditions of the magma ocean that may have occupied much or all of the mantle of the early Earth. The authors find, on the basis of first principles molecular dynamic simulations that the viscosity of the magma ocean is very low ($<0.1 \text{ Pa s}$) and that it increases monotonically with increasing pressure at high pressure. The latter result is at odds with another recent study, and the present authors do a nice job of highlighting the shortcomings of the previous study. In particular, the present authors use much longer run times and a much more careful analysis of ergodicity to demonstrate the accuracy of this results.

I recommend publication after consideration of the following points:

The authors thank the reviewer for the recommendation.

1) l. 116. "Limited available space". This is ambiguous and I don't think its what the authors mean. I think they mean "diminished free volume" or something like it.

We have modified the sentence to correct it (lines # 116-118). Thank you for your suggestion.

2) ll. 260-267. I do not completely follow the arguments here. They start in the context of the magma ocean, and end at the LAB. The problem is that the LAB is much colder than the

temperature being considered. Are the authors extrapolating their results to the conditions of the LAB in order to make this argument?

Thank you for raising this important question. Our simulation at 2200 K is not much hotter compared to the potential temperature at LAB. We have extrapolated viscosity to a lower temperature relevant for LAB^{5,6}. However, our interpretation is mostly qualitative based on the viscosity minima at ~4 GPa which is roughly 120 km depth. Since viscosity is lowest at a depth below 90 km, magma mobility is highest below LAB. For the present Earth's interior, it implies that upwelling magma tends to cumulate at a shallower depth of LAB due to increased mobility as suggested in the previous experiment⁵. We have revised the draft to address this issue (lines # 266-268).

3) l. 271. Why does basaltic melt have a lower viscosity than MgSiO₃ melt? Can some rationalization or insight be provided?

Thank you for your suggestion. Some of the discrepancies between the previous FPMD simulations of MgSiO₃ and basaltic melt can be attributed to the difference in exchange correlation functional (LDA vs GGA method). Melt compositions have a small effect on the viscosity when compared as a function of density (**Supplementary Figure S6**). We added more discussion in the revised manuscript (lines # 212-215). We have also added viscosity of MgSiO₃ melt in **Supplementary Figure S5-S6** to demonstrate this point.

4) l. 281. The authors should use instead values of the thermal conductivity of a silicate melt determined at high pressure for the first time via ab initio simulation (GRL, 2021).

Thank you for your suggestion. We have revised the values of thermal conductivity using the recent study of the thermal conductivity of silicate melt at high pressure and 4000 K⁷ and also revised Rayleigh and Prandtl numbers based on new thermal conductivity data (lines # 288-289).

5) Last paragraph. The authors relate their viscosity to the cooling time of the magma ocean via the Rayleigh number and scaling laws from the literature. This is fine as far as it goes, but should at least mention some other very important, probably controlling factors: the presence of an insulating atmosphere, and the effect of partial crystallization. Either of these factors can change the cooling time by many orders of magnitude and are probably more important considerations than a few orders of magnitude difference in melt viscosity.

Thank you for your suggestion. We have modified the discussion to address this issue (lines # 301-306).

6) ll. 296-299. This sentence does not make sense. It could be abbreviated: "If significant chemical heterogeneity is already present...then chemical heterogeneity is present."

Thank you for pointing it out. We have modified this portion of the manuscript.

7) Last paragraph. The arguments for fractional crystallization is not clearly explained. Why does low viscosity lead to fractional crystallization exactly? Large melt mobility? Large crystal settling velocity? The authors need to clarify.

We rewrote this paragraph to explain the arguments more clearly. Please refer to lines #307-330 of the revised manuscript. Thank you for your suggestions.

8) l. 335. *The authors should justify their choice of 4/3 weighting. They could show the stress auto-correlation function and the viscosity computed from each individual stress component.*

Thank you for the suggestion. We have added a supplementary figure to show the stress auto-correlation function and the viscosity of individual stress components (**Supplementary Figure S11**).

REVIEWERS' COMMENTS ROUND 2

Reviewer #1 (Remarks to the Author):

The authors have addressed the questions raised. I recommend publication of the manuscript.

Reviewer #2 (Remarks to the Author):

The authors have provided thorough responses to the comments and queries in my original review, in particular, to comment on finite-size effect and I suggest that it is now ready to be published.

Authors would like to thank the reviewers for their recommendation.

References

- 1 Dingwell, D. B. & Webb, S. L. Structural relaxation in silicate melts and non-Newtonian melt rheology in geologic processes. *Physics and Chemistry of Minerals* **16**, 508-516 (1989).
- 2 Yasuda, A., Fujii, T. & Kurita, K. Melting phase relations of an anhydrous mid-ocean ridge basalt from 3 to 20 GPa: Implications for the behavior of subducted oceanic crust in the mantle. *Journal of Geophysical Research: Solid Earth* **99**, 9401-9414 (1994).
- 3 Hirose, K., Fei, Y., Ma, Y. & Mao, H.-K. The fate of subducted basaltic crust in the Earth's lower mantle. *Nature* **397**, 53-56 (1999).
- 4 Majumdar, A., Wu, M., Pan, Y., Iitaka, T. & John, S. T. Structural dynamics of basaltic melt at mantle conditions with implications for magma oceans and superplumes. *Nature communications* **11**, 1-9 (2020).
- 5 Sakamaki, T. *et al.* Ponded melt at the boundary between the lithosphere and asthenosphere. *Nat. Geosci.* **6**, 1041-1044 (2013).
- 6 Bajgain, S. K. & Mookherjee, M. Carbon bearing aluminosilicate melt at high pressure. *Geochimica et Cosmochimica Acta* **312**, 106-123, doi:10.1016/j.gca.2021.07.039 (2021).
- 7 Deng, J. & Stixrude, L. Thermal conductivity of silicate liquid determined by machine learning potentials. *Geophysical Research Letters* **48**, e2021GL093806, doi:doi.org/10.1029/2021GL093806 (2021).